# Loss of Zonula Occludens-1 (ZO-1) Enhances Angiogenic Signaling in Ovarian Cancer Cells

**DOI:** 10.3390/ijms26178389

**Published:** 2025-08-29

**Authors:** Seongsoo Choi, Ki Hyung Kim, Min-Hye Kim, HyoJin An, Do-Ye Kim, Wan Kyu Eo, Ji Young Lee, Hongbae Kim, Heungyeol Kim, Hee-Jae Cha

**Affiliations:** 1Departments of Parasitology and Genetics, Kosin University College of Medicine, Busan 49241, Republic of Korea; css2349@gmail.com (S.C.); kmhmary93@naver.com (M.-H.K.); ppuyajin@naver.com (H.A.); 2Department of Obstetrics and Gynecology, Pusan National University Hospital, Busan 49241, Republic of Korea; ghkim@pusan.ac.kr; 3Biomedical Research Institute, Pusan National University Hospital, Busan 49241, Republic of Korea; 4College of Pharmacy, Kyungpook National University, Daegu 41566, Republic of Korea; dongkae97@gmail.com; 5Department of Internal Medicine, College of Medicine, Kyung Hee University, Seoul 02447, Republic of Korea; wkeo@yahoo.com; 6Department of Obstetrics and Gynecology, Konkuk University School of Medicine, Seoul 05029, Republic of Korea; jylee@kuh.ac.kr; 7Department of Obstetrics and Gynecology, Kangnam Sacred Heart Hospital, Hallym University Medical Center, Hallym University College of Medicine, Seoul 07440, Republic of Korea; drhbkim@unitel.co.kr; 8Department of Obstetrics and Gynecology, Hannah Hospital, Busan 48312, Republic of Korea; hykyale@gmail.com; 9Institute for Medical Science, Kosin University College of Medicine, Busan 49241, Republic of Korea

**Keywords:** ovarian cancer, ZO-1, angiogenesis, CXCL8, KLF5

## Abstract

Zonula occludens-1 (ZO-1), encoded by the TJP1 gene, is a crucial scaffolding protein within tight junctions that maintains epithelial and endothelial barrier integrity. In addition to its structural role, ZO-1 participates in signal transduction pathways that influence various cellular processes such as proliferation, differentiation, and apoptosis. Increasing evidence suggests that tight junction proteins, including ZO-1, play important regulatory roles in tumor progression, particularly by modulating metastasis, cell polarity, and vascular remodeling. Ovarian cancer, the most lethal gynecologic malignancy, is characterized by rapid growth, peritoneal dissemination, and a strong reliance on tumor angiogenesis. However, the specific role of ZO-1 in regulating angiogenesis within ovarian cancer remains poorly defined. In this study, we used CRISPR-Cas9-mediated gene editing to generate TJP1 knockout (KO) ovarian cancer cell lines and investigated the impact of ZO-1 loss on the expression of angiogenesis-related genes. Transcriptomic and qRT-PCR analyses revealed upregulation of KLF5 and IL-8, both of which are well-established pro-angiogenic factors. Furthermore, functional assessment using a Matrigel™ tube formation assay demonstrated that conditioned media from ZO-1-deficient cells significantly enhanced endothelial tube formation. These findings indicate that ZO-1 loss promotes a pro-angiogenic tumor microenvironment, likely through modulation of key signaling molecules such as KLF5 and IL-8. Therefore, ZO-1 may serve as a potential suppressor of angiogenesis and a therapeutic target in ovarian cancer.

## 1. Introduction

Tight junctions (TJs) are specialized intercellular adhesion complexes that form selective barriers in epithelial and endothelial tissues, playing key roles in maintaining tissue integrity, regulating paracellular permeability, and coordinating signaling pathways essential for cell proliferation and immune responses [1,2,3,4,5]. These structures consist of transmembrane proteins such as claudins, occludin, and junctional adhesion molecules (JAMs) anchored to the actin cytoskeleton via cytoplasmic scaffold proteins [6,7]. Among them, ZO-1, encoded by the TJP1 gene, acts as a critical adaptor linking TJ components to the cytoskeleton and various signaling molecules, thereby conferring both structural stability and regulatory plasticity [8,9].

ZO-1 not only maintains junctional architecture but also responds to mechanical and biochemical cues in the tumor microenvironment, influencing processes such as epithelial–mesenchymal transition (EMT), proliferation, and immune cell infiltration [10]. Dysregulation of ZO-1 either through downregulation or cytoplasmic mislocalization has been implicated in diverse malignancies, leading to compromised cell adhesion, increased motility, and enhanced invasiveness [11,12]. Such alterations are associated with poor prognosis and metastasis in multiple cancers, including those of the liver, stomach, pancreas, and lung [13,14,15,16].

In ovarian cancer tissues, ZO-1 expression is frequently downregulated or redistributed from the plasma membrane to the cytoplasm or nucleus. This alteration has been associated with EMT and enhanced tumor invasiveness [17,18]. Ovarian cancer is the most lethal gynecologic malignancy, characterized by asymptomatic progression, frequent relapse, and a tendency for peritoneal dissemination [18,19]. These metastatic features suggest a role for adhesion molecule dysregulation, including TJ disassembly. While emerging evidence implicates ZO-1 in modulating the tumor microenvironment through inflammatory and stromal interactions [17], its influence on tumor angiogenesis in ovarian cancer has yet to be elucidated.

Angiogenesis is a hallmark of cancer that enables tumor growth and metastasis by providing oxygen, nutrients, and routes for dissemination [20,21]. It is controlled by a balance between pro- and anti-angiogenic signals, and its disruption leads to abnormal tumor vasculature [22]. Recent studies suggest that junctional proteins like ZO-1 may intersect with angiogenic signaling, either directly through gene regulation or indirectly via microenvironmental modulation [23,24,25]. However, the mechanistic relationship between ZO-1 and angiogenesis remains largely unknown in ovarian cancer.

In this study, we aim to elucidate the role of TJP1 (ZO-1) in regulating tumor angiogenesis in ovarian cancer. Using SKOV3 and OVCAR3 cell lines as models, we generated CRISPR-Cas9-mediated TJP1 knockout cells and assessed angiogenic behavior through both in vitro tube formation assays and in vivo Matrigel plug experiments. To further investigate the molecular pathways affected by ZO-1 loss, we examined the expression of key angiogenic mediators, including IL-8 and KLF5. This study seeks to provide new insights into the functional relevance of ZO-1 in ovarian cancer pathogenesis and highlight its potential as a target for anti-angiogenic therapeutic strategies.

## 2. Results

### 2.1. Establishment of ZO-1-Deficient Ovarian Cancer Cell Lines

To investigate the functional role of ZO-1 in ovarian cancer, we generated ZO-1 KO cell lines using the CRISPR/Cas9 genome editing system. Two epithelial ovarian cancer cell lines, SKOV3 and OVCAR3, were transfected with plasmids encoding Cas9 and guide RNAs (gRNAs) specifically targeting the ZO-1 gene (TJP1). The gRNA sequences used for targeting are listed in Figure 1A. To confirm successful genome editing, the targeted genomic regions were amplified and subjected to Sanger sequencing. As shown in Figure 1B, sequence analysis revealed the presence of deletions at the target loci, resulting in frameshift mutations and subsequent alterations in the encoded amino acid sequence. These findings confirm that the CRISPR/Cas9 system efficiently introduced disruptive mutations into the TJP1 gene. Subsequently, we assessed ZO-1 protein expression in the edited cells by Western blot analysis. As presented in Figure 1C, ZO-1 protein was undetectable in both SKOV3 and OVCAR3 KO cell lines, indicating a complete loss of ZO-1 protein expression. Together, these results verify the successful establishment of ZO-1-deficient ovarian cancer cell lines, which serve as a robust model to explore the biological role of ZO-1 in tumor progression.

### 2.2. Enhanced in Vitro Tumor Angiogenesis Following ZO-1 Knockout

To evaluate the impact of ZO-1 loss on tumor angiogenesis, we performed an in vitro tube formation assay using conditioned media (CM) derived from ZO-1 KO ovarian cancer cells. CM was collected from SKOV3 and OVCAR3 cells following CRISPR/Cas9-mediated ZO-1 depletion. In parallel, CM was also obtained from ZO-1-rescued cell lines to determine whether re-expression of ZO-1 could reverse the pro-angiogenic phenotype (Appendix A). Human umbilical vein endothelial cells (HUVECs) were seeded onto Matrigel-coated 96-well plates and treated with the collected CM for six hours. As shown in Figure 2A,B, CM from ZO-1 KO SKOV3 and OVCAR3 cells significantly increased endothelial tube formation compared to CM from control cells. Notably, the pro-angiogenic effect was attenuated when ZO-1 expression was restored in the KO cells, indicating that ZO-1 negatively regulates angiogenic potential in ovarian cancer cells. To quantitatively assess angiogenesis, tube network parameters such as mesh length and segment length were measured and are presented in Figure 2C,D. These analyses further confirmed that ZO-1 depletion leads to enhanced vascular network formation, which is reversed upon ZO-1 re-expression. Collectively, these findings demonstrate that ZO-1 plays a suppressive role in the regulation of angiogenesis in ovarian cancer, and its loss promotes a tumor microenvironment conducive to vascularization.

### 2.3. ZO-1 Loss Induces Angiogenesis in the Matrigel Plug Assay

To evaluate the role of ZO-1 in tumor-induced angiogenesis, a Matrigel plug assay was performed in BALB/c nude mice. SKOV3 ovarian cancer cells were suspended in Matrigel and subcutaneously injected into the flanks of the mice (Appendix A). Two groups were established: a control (CON) group using wild-type SKOV3 cells and a ZO-1 KO group using ZO-1-deficient SKOV3 cells (Figure 3A). The mice were maintained for two weeks to allow tumor cells to interact with the surrounding microenvironment and induce vascularization within the plugs. After two weeks, the Matrigel plugs were harvested and assessed for gross morphology. Compared to the CON group, plugs from the ZO-1 KO group appeared larger and exhibited more prominent surface vascularization, as indicated by visible blood vessels (Figure 3B, red arrows). To assess vascularization at the histological level, the plugs were fixed, embedded in paraffin, sectioned, and subjected to hematoxylin and eosin (H&E) staining. Histological analysis revealed a significantly higher number of blood vessels in the ZO-1 KO group compared to the CON group (Figure 3C, red arrows). Quantification of vascular density demonstrated a marked increase in blood vessel formation in the ZO-1 KO plugs relative to controls (Figure 3C, bar graph; *p* < 0.01). These results suggest that loss of ZO-1 promotes angiogenesis, potentially by modulating the tumor microenvironment to favor neovascularization.

### 2.4. Regulation of ZO-1–Dependent Angiogenesis by IL-8 and KLF5

Based on the results described above, ZO-1 appears to play a regulatory role in angiogenesis in ovarian cancer cells. To further elucidate the underlying molecular mechanisms, transcriptome analysis was performed using RNA sequencing. Gene expression changes associated with angiogenesis were analyzed in both SKOV3 and OVCAR3 ovarian cancer cell lines. Figure 4A shows that 17 angiogenesis-related genes were differentially expressed in SKOV3 cells following ZO-1 KO, with 13 genes upregulated and 4 downregulated. In OVCAR3 cells, 56 genes exhibited significant differential expression, including 22 upregulated and 34 downregulated genes. Additional candidate genes potentially involved in angiogenesis were identified through heatmap analysis of differentially expressed angiogenesis-related genes (Figure 4B). Comparison of these transcriptomic results with tube formation assay data revealed commonly upregulated genes in both cell lines, notably Krüppel-like factor 5 (KLF5) and C-X-C motif chemokine ligand 8 (CXCL8/IL-8). IL-8 has previously been implicated in promoting angiogenesis in pancreatic cancer, while KLF5 has been reported to enhance angiogenic activity in bladder cancer [26,27]. To validate these findings, quantitative real-time PCR (qRT-PCR) was conducted and confirmed that IL-8 and KLF5 mRNA levels were significantly elevated in ZO-1 KO cells compared to controls. Specifically, IL-8 expression increased 2.39-fold in SKOV3 cells and 19.48-fold in OVCAR3 cells, while KLF5 expression showed a 1.77-fold change in SKOV3 cells and a 1.87-fold increase in OVCAR3 cells (Figure 4C). Consistently, Western blot analysis further demonstrated increased protein expression levels of IL-8 and KLF5 in both SKOV3 and OVCAR3 cells following ZO-1 deletion (Figure 4D). Densitometric quantification of the immunoblot signals confirmed these findings, showing a significant upregulation of IL-8 and KLF5 compared with control cells. Collectively, these results suggest that IL-8 and KLF5 are key mediators of ZO-1–regulated angiogenesis in ovarian cancer cells.

## 3. Discussion

In this study, we demonstrated that CRISPR-Cas9-mediated knockout of ZO-1 in human ovarian cancer cell lines (SKOV3 and OVCAR3) enhances angiogenic potential. Functional assays revealed that conditioned media from ZO-1-deficient cells promoted tube formation in HUVECs, indicating that ZO-1 may regulate the secretion of pro-angiogenic factors. This phenotype was reversed by re-expression of ZO-1, suggesting a ZO-1–dependent mechanism that modulates endothelial behavior. Given that tumor cells do not form blood vessels themselves [28,29], these results support a role for ZO-1 in regulating the tumor microenvironment by influencing the behavior of neighboring endothelial cells rather than acting solely within tumor cells.

Consistent with in vitro findings, Matrigel plug assays in vivo showed significantly increased vascularization in plugs containing ZO-1 knockout cells, reinforcing the anti-angiogenic role of ZO-1.

Transcriptomic and qRT-PCR analyses identified IL-8 and KLF5 as key pro-angiogenic genes upregulated in ZO-1–deficient cells. IL-8 is a chemokine known to promote endothelial proliferation, migration, and vascular remodeling [30,31,32], while KLF5 is a zinc finger transcription factor involved in angiogenesis and tumor progression through regulation of multiple downstream targets [26,33,34]. These findings suggest that loss of ZO-1 enhances angiogenesis by upregulating IL-8 and KLF5.

Beyond its structural role at tight junctions, ZO-1 has also been implicated in regulating gene expression and transcription factor activity in epithelial cells. For example, ZO-1 can interact with the Y-box transcription factor ZONAB/YBX3, thereby controlling cell cycle regulators such as cyclin D1 [35], and with symplekin to influence transcriptional programs [36]. These precedents suggest that the upregulation of IL-8 and KLF5 observed in ZO-1-deficient ovarian cancer cells may reflect a broader function of ZO-1 in modulating transcriptional activity, linking tight junction integrity to gene expression pathways that drive angiogenesis.

Interestingly, previous studies have proposed that IL-8 overexpression can suppress ZO-1 expression, indicating a reciprocal regulatory loop between ZO-1 and angiogenic signaling pathways [37,38]. Moreover, ZO-1 has been proposed to function as a tumor suppressor through regulation of the tumor microenvironment and angiogenic signaling [39,40,41].

While our data strongly support a role for ZO-1 as a negative regulator of tumor angiogenesis, further studies are warranted to determine whether inhibition of IL-8 or KLF5 can reverse the angiogenic phenotype in ZO-1–deficient cells, and whether ZO-1 loss contributes to tumor progression and metastasis in vivo.

In conclusion, our findings reveal that ZO-1 suppresses angiogenesis in ovarian cancer by negatively regulating pro-angiogenic mediators such as IL-8 and KLF5. This highlights a novel role for ZO-1 in modulating the tumor microenvironment and suggests that restoring ZO-1 function or targeting its downstream effectors may represent promising strategies for limiting angiogenesis and disease progression in ovarian cancer.

## 4. Materials and Methods

### 4.1. Cell Culture

Human ovarian cancer cell lines SKOV3 and OVCAR3 were obtained from the Korean Cell Line Bank (KCLB, Seoul, Republic of Korea), and HUVECs were purchased from ATCC (Manassas, VA, USA). SKOV3 and OVCAR3 cells were cultured in RPMI-1640 medium supplemented with 10% fetal bovine serum (FBS), 1% penicillin/streptomycin, and L-glutamine. HUVECs were cultured in M199 medium containing 20% FBS, 1% Antibiotic-Antimycotic (Gibco, Thermo Fisher Scientific, Waltham, MA, USA), and 1% endothelial cell growth supplement (ECGS). All cells were maintained at 37 °C in a humidified incubator with 5% CO_2_.

### 4.2. Western Blot Analysis

Proteins were extracted using PRO-PREP™ (iNtRON Biotechnology, Seongnam, Republic of Korea), separated by SDS-PAGE on a Bolt™ 4–12% Bis-Tris gel (Invitrogen, Thermo Fisher Scientific, Waltham, MA, USA), and transferred onto nitrocellulose membranes. Membranes were blocked in TBS-T containing 5% skim milk for 2 h at room temperature and incubated overnight at 4 °C with primary antibodies: ZO-1 (1:1000, Invitrogen), IL-8 (1:1000, Abcam, Cambridge, UK ), KLF5 (1:1000, Abclonal, Woburn, MA, USA), and GAPDH (1:5000, Invitrogen). After washing, membranes were incubated with HRP-conjugated secondary antibodies, and signals were detected using an ECL detection kit.

### 4.3. Generation of ZO-1 Knockdown Cells

CRISPR-Cas9-mediated knockout of ZO-1 was performed using gRNA (5′-ACATACAGTGACGCTTCACA-3′) designed via the online tool of Bioneer (Daejeon, Republic of Korea). The gRNA was cloned into pRGEN-Cas9-CMV/T7-Hygro-EGFP and CRISPR/Cas9-Puro plasmids (ToolGen, Seoul, Republic of Korea). SKOV3 and OVCAR3 cells were transfected and selected with 100 µg/mL hygromycin for 48 h. After two weeks, individual clones were isolated using cloning cylinders and validated by Sanger sequencing and Western blotting.

### 4.4. Tube Formation Assay

Matrigel (Corning, NY, USA) was added to 96-well plates and polymerized at 37 °C for 1 h. HUVECs were suspended in 100 µL of conditioned media (CM) and seeded onto the Matrigel. After 6–8 h incubation, tube structures were examined using a bright-field microscope (Carl ZEISS Axio Observer 7, Carl Zeiss, Germany), and images were acquired with ZEN 3.8 software (Carl Zeiss, Oberkochen, Germany). No fluorescent probes were used in this assay. Tube formation parameters, including total tube length, number of meshes, and segments, were quantified from the captured images using ImageJ software, version 1.54g (NIH, Bethesda, MD, USA).

### 4.5. Matrigel Plug Assay

Matrigel was thawed overnight at 4 °C before use. SKOV3 and ZO-1 KO SKOV3 cells (1 × 10^6^) were mixed with 250 µL of Matrigel and 50 µL of medium (total 300 µL) and then injected subcutaneously into the flanks of 6–8-week-old BALB/c nude mice (*n* = 5 per group). Mice were maintained under specific pathogen-free (SPF) conditions with free access to food and water and a 12 h light/dark cycle. After 14 days, mice were sacrificed under anesthesia, and plugs were fixed in 4% paraformaldehyde, paraffin-embedded, and sectioned for histological analysis. All animal experiments were conducted in accordance with the Animal Protection Act of Korea and the Laboratory Animal Act of Korea, as well as the ARRIVE guidelines and the Guide for the Care and Use of Laboratory Animals (NIH, 8th edition). The experimental protocol was reviewed and approved by the Institutional Animal Care and Use Committee (IACUC) of Kosin University College of Medicine (Approval No. KUCMIACUC [KMAP-24-09]).

### 4.6. Collection of Conditioned Media

SKOV3, OVCAR3, and their respective ZO-1 KO cells were cultured in 6-well plates. Upon confluency, cells were incubated in serum-free medium for 4–6 h, followed by replacement with medium containing 10% FBS. Conditioned media were collected 24 h later, centrifuged to remove debris, and used for downstream assays. The conditioned media (CM) were pooled and stored at −20 °C for up to one week. All CM experiments were performed using immortalized ovarian cancer cells within 20 passages after thawing

### 4.7. Quantitative Real-Time PCR (qRT-PCR)

Total RNA was extracted using TRIzol Reagent (Invitrogen), and cDNA was synthesized using a reverse transcription kit (Bioneer, Daejeon, Republic of Korea). qRT-PCR was conducted using TB Green Premix Taq (Takara, Kusatsu, Shiga, Japan) on a QuantStudio 3 system (Thermo Fisher Scientific). Relative expression was calculated by the 2^−ΔΔCt^ method and normalized to *GAPDH*. The primer sequences for *CXCL8* genes are as follows: *CXCL8* sense, 5′-CAG TTT TGC CAA GGA GTG CT-3′; antisense, 5′-ACT TCT CCA CAA CCC TCT GC-3′; KLF5 sense, 5′-ATT TAA AAG CTC ACC TGA GGA C-3′; antisense, 5′-CTG GTG CCT CTT CAT ATG C-3. *GAPDH* was used as a control (sense primer, 5′-CAA TGA CCC CTT CAT TGA CC-3′; antisense primer, 5′-GAC AAG CTT CCC GTT CTC AG-3′).

### 4.8. Hematoxylin and Eosin (H&E) Staining

Paraffin-embedded sections were deparaffinized in xylene, rehydrated through graded ethanol, and stained with hematoxylin (3–5 min) and eosin (1 min). Slides were dehydrated, cleared in xylene, and mounted using synthetic mounting medium. H&E-stained sections were examined using a bright-field microscope (Nikon ECLIPSE 50i, Nikon, Tokyo, Japan) equipped with a digital camera. Images were acquired using NIS-Elements software version 4.0 (Nikon, Japan). Quantification of blood vessel density was performed from at least three randomly selected fields per section.

### 4.9. mRNA Sequencing and Analysis

Transcriptomic profiling was performed by Ebiogen Inc. (Seoul, Republic of Korea). Total RNA from SKOV3, ZO-1 KO SKOV3, OVCAR3, and ZO-1 KO OVCAR3 cells was subjected to next-generation sequencing. Differentially expressed genes (DEGs) were identified, and cluster analysis and visualization were conducted using ExDEGA software (v5.1.1.4, Ebiogen).

### 4.10. Plasmid Construct for Over-Expression

The plasmid construct used in this study was the pcDNA3.1+/C-(K)-DYK expression vector. The Tjp1_OMu20010 plasmid (catalogue no. U9042HE240-1) was purchased from GenScript (Piscataway, NJ, USA). Plasmid transfection into ZO-1 knockout SKOV3 and OVCAR3 ovarian cancer cells was performed using Lipofectamine 2000 (Thermo Fisher Scientific, Waltham, MA, USA) according to the manufacturer’s instructions.

### 4.11. Statistical Analysis

All in vitro experiments were performed in triplicate. Statistical significance was evaluated using Student’s t-test in GraphPad Prism 8.0.2 (GraphPad Software, San Diego, CA, USA). Data are presented as mean ± SD, and *p*-values were considered significant as follows: *p* < 0.05, *p* < 0.01, and *p* < 0.001.

## Figures and Tables

**Figure 1 ijms-26-08389-f001:**
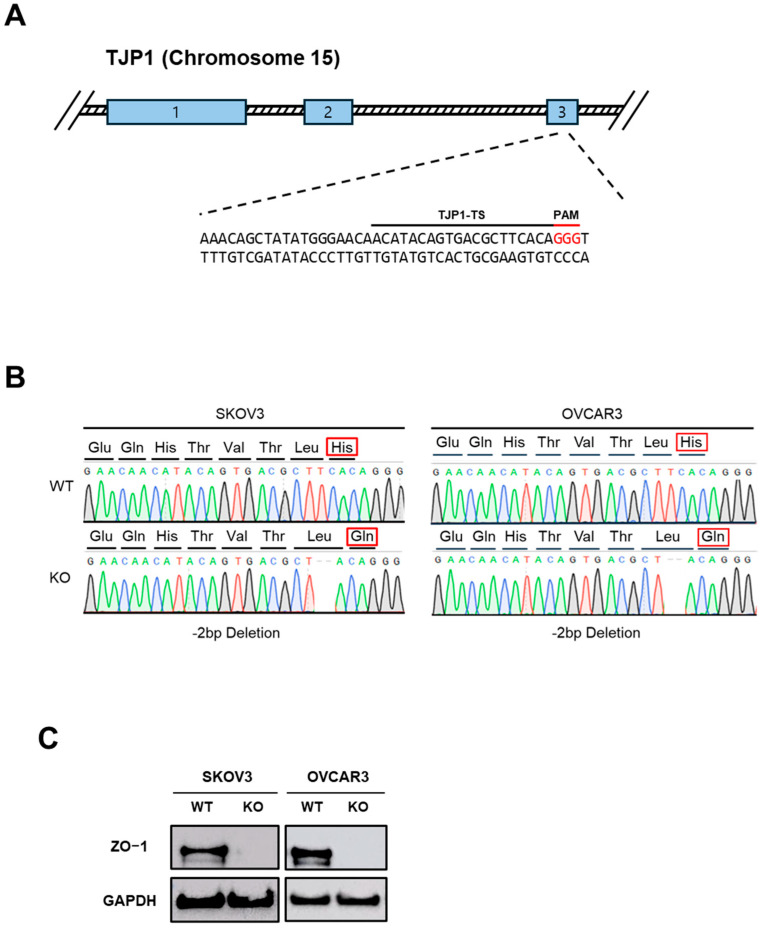
(**A**) Exon 3 of the TJP1 gene, encoding ZO-1, was targeted in SKOV3 and OVCAR3 ovarian cancer cell lines using the CRISPR-Cas9 system. (**B**) Successful genome editing and suppression of ZO-1 expression were confirmed by Sanger sequencing. (**C**) ZO-1 protein depletion was further validated by Western blot analysis.

**Figure 2 ijms-26-08389-f002:**
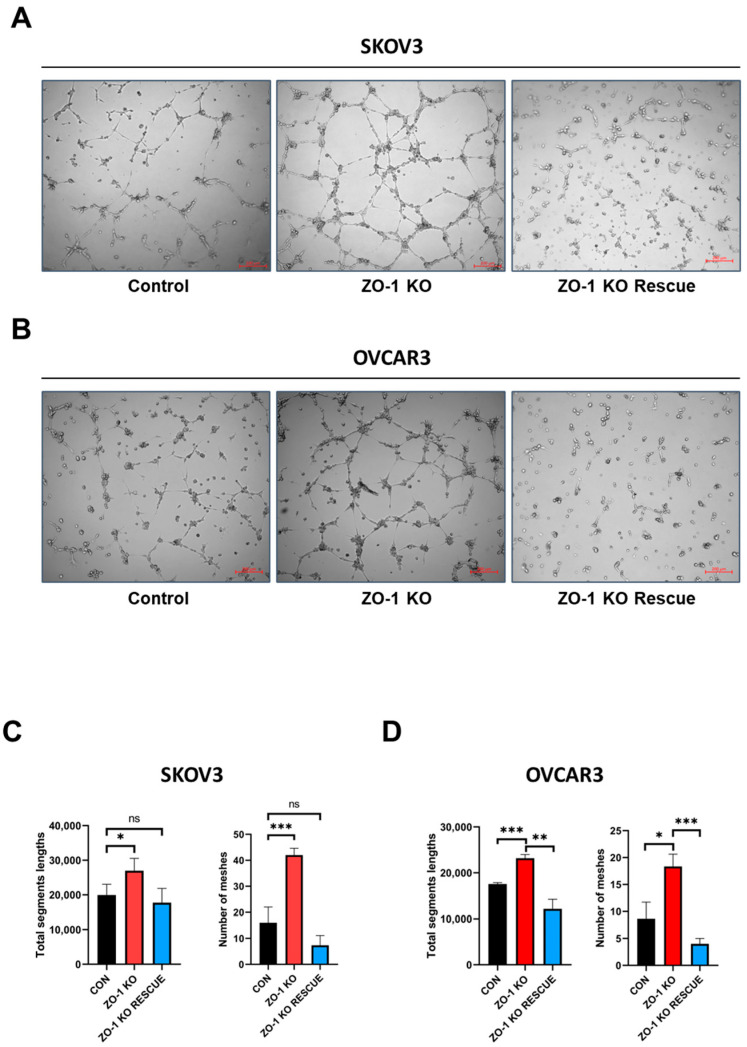
Verification of the role of ZO-1 on angiogenesis in human ovarian cancer cells. (**A**,**B**) Tube formation assays were performed using HUVECs incubated with conditioned medium (CM) derived from ZO-1 KO SKOV3 and OVCAR3 human ovarian cancer cells. Tube formation assays using conditioned medium (CM) from ZO-1 KO SKOV3 and OVCAR3 cells showed a significant increase in tube formation in HUVECs. (**C**,**D**) This effect was reversed when ZO-1 expression was rescued in ZO-1 KO SKOV3 and OVCAR3 cells. Scale bar = 200 µm. The above results were quantified and represented graphically for statistical analysis. (* *p* < 0.05, ** *p* < 0.01, *** *p* < 0.001).

**Figure 3 ijms-26-08389-f003:**
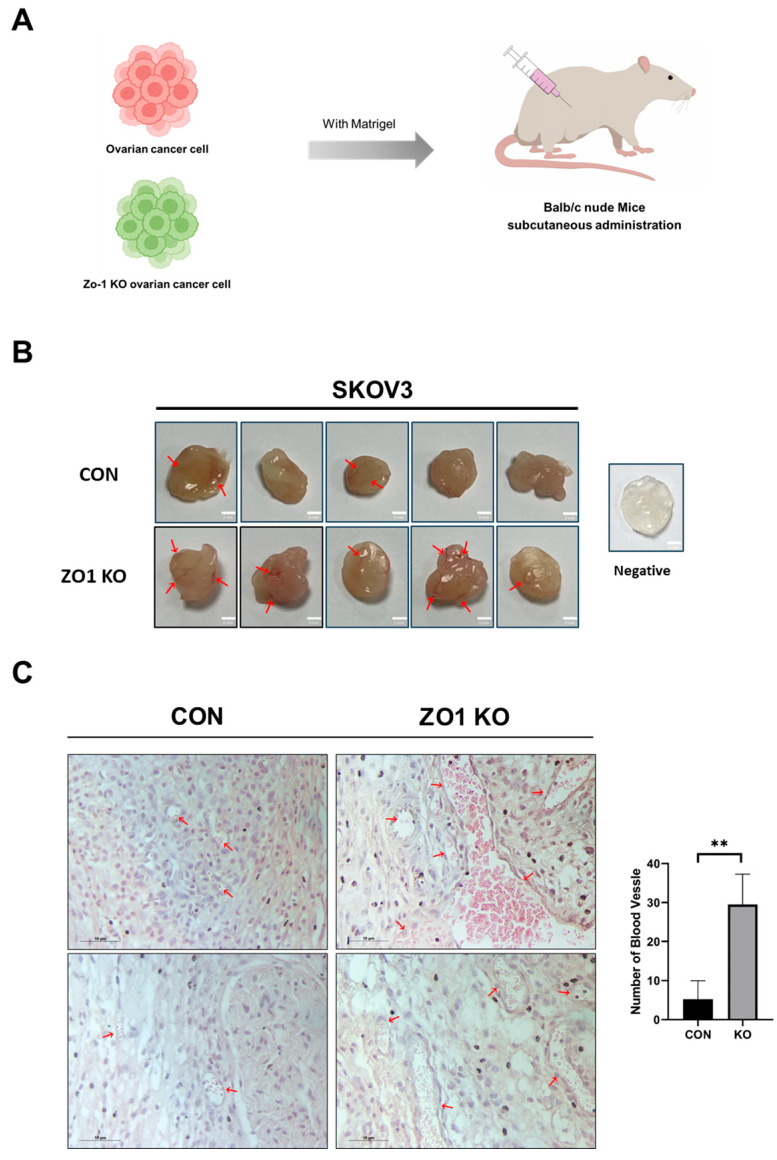
ZO-1 deficiency enhances blood vessel formation in Matrigel. (**A**) Schematic diagram of the experimental procedure. Control or ZO-1 KO SKOV3 ovarian cancer cells were mixed with Matrigel and subcutaneously injected into BALB/c nude mice (*n* = 5 per group) to evaluate the angiogenic potential in vivo. (**B**) Representative images of Matrigel plugs retrieved two weeks after injection. Red arrows indicate visible blood vessel infiltration within the plugs. The ZO-1 KO group showed a notable increase in vascularization compared to the control (CON) group. (**C**) Hematoxylin and eosin (H&E) staining of paraffin-embedded Matrigel plugs. Red arrows indicate blood vessels. Quantification of blood vessel numbers per field was performed (right graph). Data are presented as mean ± SD (*n* = 5). ** *p* < 0.01, unpaired Student’s *t*-test.

**Figure 4 ijms-26-08389-f004:**
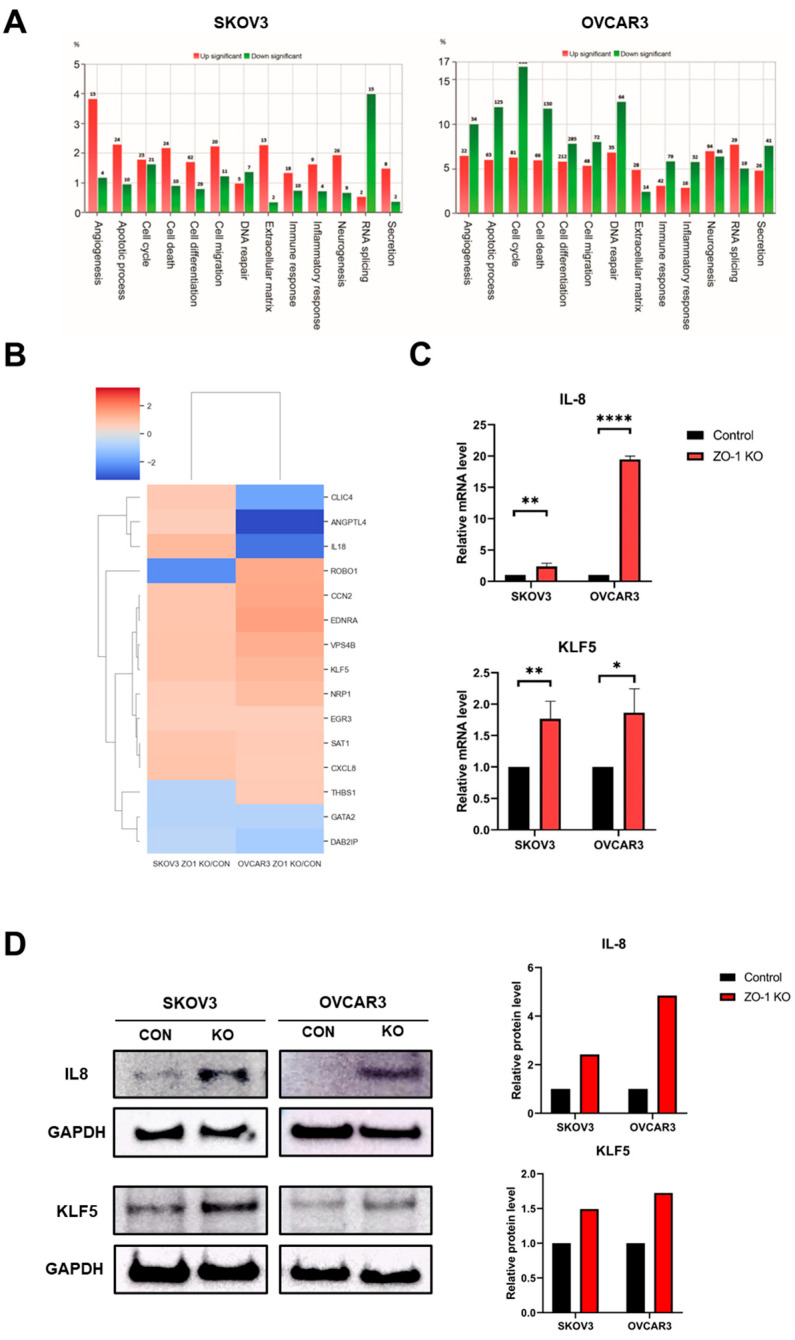
Transcriptomic and molecular analyses reveal upregulation of angiogenesis-related genes following ZO-1 KO in ovarian cancer cells. (**A**) Gene ontology (GO) analysis of differentially expressed genes (DEGs) in TJP1 (ZO-1) KO cells was performed using next-generation RNA sequencing. Enriched biological processes associated with angiogenesis and cell signaling are presented for SKOV3 and OVCAR3 cells. (**B**) Heatmap visualization of angiogenesis-related gene expression across control and ZO-1 KO groups in SKOV3 and OVCAR3 cells. Columns represent experimental groups, and rows represent individual angiogenesis-related genes. (**C**) Quantitative real-time PCR (qRT-PCR) analysis of CXCL8 (IL-8) and KLF5 mRNA expression in SKOV3 and OVCAR3 cells. Gene expression was normalized to GAPDH. Data are presented as mean ± SD (*n* = 3). * *p* < 0.05, ** *p* < 0.01, **** *p* < 0.0001, compared to control. (**D**) Western blot analysis of IL-8 and KLF5 protein expression in control and ZO-1 KO SKOV3 and OVCAR3 cells. GAPDH was used as a loading control. Densitometric quantification of the protein expression levels is shown on the right. (** *p* < 0.01, *** *p* < 0.001).

## Data Availability

All data are contained in this article and there are no repository data.

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
