# Peer review of "Loss of Zonula Occludens-1 (ZO-1) Enhances Angiogenic Signaling in Ovarian Cancer Cells"

_ijms, 2025, doi:10.3390/ijms26178389_

Round 1
Reviewer 1 Report
Comments and Suggestions for Authors
The work is well planned and executed and the manuscript presentation overall is of good quality, however important elements are not included and have to be addressed by the authors.
- The word "enhances" in the title is spelled incorrectly as "eenhances".
- Please define the abbreviation JAMs when first used in line 45 in the Introduction.
- The authors refer to supplementary material and figures in the manuscript, but these data are not included. The additional file shows the raw blot images only. Line 294 indicates that supplementary materials can be downloaded but no is link provided. This is crucial to complete the review of the manuscript.
- Please add scale bars to the images presented in Figure 3.
- Please report the fold differences for the mRNA data reported in lines 173-175.
- The discussion section requires more substantial arguments, e.g. the authors identify IL-8 and KLF5 among 17 angiogenesis-related genes. Here the authors must substantiate why these two genes were targeted for further analysis. Additional inhibitory experiments of IL-8 and KLF5 to show a causal link in the ZO-1 KO model, may in fact strengthen the argument and the manuscript overall.
- Several methodological omissions should be included, as indicated below:
- Line 250 states that clones were validated by RT-PCR, qRT-PCR and Western blotting - please show data.
- Line 255 states that the tube structures were evaluated under a fluorescence microscope. Please include the instrument details, along with Ex/Em wavelenghths and which probes were included in the assay. Additionally, please clarify how the data analyses were performed.
- Please include ethics information and animal care and use statements for the Matrigel plug assays performed on the BALB/c nude mice - paragraph 4.5. This is crucial missing information and raises ethical questions.
- Did the authors store the conditioned media mentioned in line 266? Please indicate how. Also indicate the passage range used for the CM experiments.
- Include information about the H&E imaging.
Author Response
“Please find our detailed point-by-point responses in the attached document.”

Reviewer 2 Report
Comments and Suggestions for Authors
This is an interesting manuscript that implicates an important tight junction protein, ZO-1 in the regulation of proangiogenic factor secretion by ovarian cancer cells. The authors performed CRISPR/Cas9-mediated knockout of ZO-1 in ovarian cancer cell lines and demonstrated that knockout cells produce and secrete factors that promote angiogenesis in vitro and in vivo. They identified IL8 and KLF5 as potential candidates. The study has scientific merit and is well designed. The manuscript is well written. Nevertheless, a few issues need to be addressed in order to increase potential impact of this study.
Comments:
- For the in vivo study presented in Figure 2, the authors need to quantify the size of tumor xenografts to demonstrate that the observed differences in tumor vascularization do not simply reflect differences in tumor growth between control and ZO-1 knockout cells.
- Figure 4. The quality of IL8 immunoblot in OVCAR cells is poor. The blot needs to be replaced. Densitometric quantification of the protein expression in Figure 4D need to be added.
- Since ZO_1 protein family has three different members with possible interconnected functions, I wonder if loss pf ZO-1 has effect on expression of ZO-2 and ZO-3 in ovarian cancer cells.
- Discussion is shallow. The authors need to discuss available data that implicate ZO-1 in regulating gene expression and transcription factor activity in epithelial cells.
- Description of ZO-1 rescue experiment needs to be added to Materials and Methods.
Author Response

(The authors gave the same response as above.)

Round 2
Reviewer 1 Report
Comments and Suggestions for Authors
Dear Authors,
Many thanks for addressing the comments and improving the quality of the manuscript.
Reviewer 2 Report
Comments and Suggestions for Authors
Authors did a great job in addressing my comments and questions. U have nothing to add.